**Data Availability Statement:** All relevant data are within the paper and Supporting information files.

**Funding:** No financial support was obtained for this study.

# The impact of Chinese adolescents visual art participation on self-efficacy: A serial mediating role of cognition and emotion

**Genman Deer**[1], **Endale Tadesse**[2], **Zhihan Chen**[3], **Sabika Khalid**[2], **Chunhai Gao**[4]*

**1** Faculty of Education and Human Development, The Education University of Hong Kong, Hong Kong, China, **2** College of Teacher Education, Zhejiang Normal University, Zhejiang, China, **3** Center for Studies of Education and Psychology of Ethnic Minorities in Southwest China, Southwest University, Chongqing, China, **4** Faculty of Education, Shenzhen University, Shenzhen, China

* chunhaigao@hotmail.com

## Abstract

A large volume of evidence indicates that only high-class students attend extracurricular activities (Art, music, sport, dancing). On the other hand, this evidence intensively underlines the substantial importance of such extracurricular activities, particularly in visual art, in promoting children's cognitive and non-cognitive well-being. Adolescents' participation in visual art was always interrelated with enhancing their emotional affection towards the Art and cognitive skill in making one, which ultimately built solid efficacy that allows them to interact with their society. The present cross-sectional study sought to shed light on the potential impact of visual art on adolescents' emotional, cognition, and self-efficacy development, which needs to be improved in the Chinese context. Hence, randomly sampled (N = 2139) junior secondary school students were recruited from the rural province of Guizhou in Southwest China to attain the aim of the study. The study's finding affirms that students engaged in artistic activities start to develop a habit of communicating with their peers, showing their work, and commenting on works made by their peers or observed in art exhibitions or museums; such a process makes them self-efficacious. Ultimately, this paper extends the application of visual art activities from educational benefits to nonacademic development, which are the primary agents for children's well-being.

## Introduction

It is evident from the data that the visual arts are valued for their distinct intrinsic contributions: Students develop an ability to transform their ideas, images, and feelings into an art form [1–4], refine their awareness of aesthetic qualities [5]; find connections between art and culture; and cultivate dispositional outcomes such as imagination, exploration, and multiple perspectives [6–9]. A large body of evidence documented that Children or student participation in extracurricular activities and parental support are associated with children's psychological, social, physical, and academic well-being [10–12]. Indeed, children's Participation in art explains an engagement in systematically designed visual arts classes that promote the cognitive and non-cognitive skills that lay a foundation for children's future well-being.

**Competing interests:** The authors have declared that no competing interests exist.

Mainly, for students who participate in such extracurricular activities, art education which comprises painting, drawing, and sculpture, is notably considered the most effective means of engaging children to enhance their creativity, self-esteem, communication, cognition, and self-efficacy [13,14–18]. However, some studies in China discuss children involved in extracurricular activities in light of the country's attachment to aesthetic culture. Hence, discussing out-of-school extracurricular activities' influence on children's cognitive and non-cognitive development took the attention of scholars the present study sought to investigate.

Furthermore, it must be stressed that besides the cognitive development of children through art education, art education also strengthens the non-cognitive abilities (self-esteem, self-concept, self-efficacy, peer-relationship, emotions), which cannot be enhanced by other teaching instruction [19–22]. That makes art education have a definite purpose for children to promote their imagination and creativity that initiate cognition and emotion [2,9,12]. Studies claim that children being involved in extracurricular activities makes them astonishingly creative and mature compared to their peers who have not participated [18,23]. Thus, teachers should make the interaction between artistic activities and students so that the learners show a tangible emotion and cognition towards the art [1,24]. In the view of a shred of studies regarding the application of art education, less is known about the concept of art cognition and emotion. In the present study, art cognition refers to the solid artistic knowledge acquired by children through the intensive art participation they had in and out of the home. In the same light, the current study speculates that children obtain a genuine emotional attachment to visual artistic work, which emerges from extensive knowledge about different visual art values. This art emotion elucidates the expression of in-depth feelings according to the setting and looks of visual artworks children observed from their friends, museum, or art galleries. In the light of such a well-being personality built through art education, children start developing confidence in themselves for possessing a healthy cognition and emotional connection with visual artworks, called self-efficacy.

Accordingly, children who attended those art activities have shown a substantial academic and psychological development than those who did not [7,19–21,25]. A survey study revealed that children participating in art-based activities would likely foster their self-esteem, peer relationship, and life satisfaction [20,21,26,27]. Although children's art participation is significantly dependent on a socioeconomic level, it is noticed that disadvantaged children benefit more than wealthier family children [2,19,28–30]. The current study shed light on the association between parental SES and children's likelihood to participate in art education and its impact on children's psychological well-being (Emotion, Cognition, Self-efficacy).

## Art education

### Art participation and socioeconomic status

Over the decades, a great deal of literature mentioned that the likelihood of Chinese children participating in aesthetic activities depends on the genuine parental support that emerges from their social class rank [7,14,43,45]. It is certain that most public schools exclude art education from the core curriculum or do not consider it as one of the compulsory subjects. The Ministry of Education or government hesitates to invest in art education parents who want their children to be engaged in art education are mandated to take them to art schools outside the school, which demands a fortune [5,22,44,46]. Parental education is directly associated with children's likelihood of participating in art-based activities, especially outside of school, known for high-quality visual art teaching with hefty fees [48]. Likewise, a concurrent finding reflected in a longitudinal study in the USA public these scenarios reflected that students who participated in aesthetic activities had not shown a more significant academic performance than

those who did not [5]. Hence, private art schools create inequality in accessing such extracurricular activities, leading to children's self-esteem, creativity, peer-relationship, motivation, and disparity in life satisfaction [14,43,47]. Emerging evidence noted that children from higher SES exhibit an exceeding creative ability than low SES counterparts [18]. Children from high SES families possess out-of-school extracurricular activities and parental support at home by facilitating them with the required equipment and praising their children's artwork. At the same time, low-SES parents have more extended working hours, so they have financial and time constraints [44]. On the other hand, a growing body of evidence stated that although extracurricular activities have no substantial benefit for children from a higher class, they extensively welfare children from low SES families who got the chance to participate in these activities [23,29, 47].

## Art participation, cognition, emotion, and self-efficacy

Visual art was always interrelated with an individual's emotional and cognition accounts [31–35]. Social-cognitive theorists conceive an individual's personality as a cognitive-affective system resulting from the concerted action of functionally distinct structures, which gradually occur throughout the development [36] cited in [37–39]. Subsequently, Bandura stated that individuals would not be involved or participate in activities without any incentive unless they dedicate themselves to attaining an intended goal [36,40]. Similarly, one belief in his/her capability on a given skill is the primary factor for individual self-efficacy to accomplish a given task for a specific and intended outcome [17,38]. The theory is widely studied to examine learning goals through personal emotion, social, behavioral, and cognition dimensions [8,26,33,41]. Identifying those activities which motivate us to be engaged to have a genuine emotional attachment is a hidden fact for many of us [31,35,39]. Likewise, parents support their children to attend art-related activities to foster their combination skills with their mates, cognition towards different artistic works, and affection or emotion towards artistical creativity, which finally leads to decent art self-efficacy [42–44] sense of achievement [21]. Alessandri and colleagues indicate that whenever children participate in art activities, they start sharing their work and praising others for the kind of painting or drawing they have made; here, cognition and emotion of art emerge, and this whole affection results in a solid self-efficacy or confidence while they take part in such art activities [21,34,37,39]. Subsequently, children who have participated in art-based activities develop personal qualities which motivate and engage them in their academic activities [7,19,27,39].

Nevertheless, such art emotion enables individuals to appraise other artistic work as long as they build a solid knowledge of cognition artists [2,31,34]. A large body of evidence claims that art cognition notably affects people's emotional states [33,42]. Physiologically emotion and cognition are centered and controlled by the nervous or brain system [33].

Accordingly, self-efficacy requires individual steadiness to possess positive behavior, which has a promising art cognition that gives him/her the confidence to praise others [2,13,38]. On the other hand, self-efficacy's effect on an individual emotional state or behavior is more indirect [8,38]. The relationship between children's art participation, cognition, emotion, and self-efficacy in aesthetic activities has yet to be fully addressed or disclosed, and few studies have tried to unfold that [16,17,35,42]. A quasi-experimental study reflects that students boost their self-efficacy through a classroom culture of giving and taking comments among students [15]. Bandura stated that those with solid self-efficacious exhibit low negative emotion even though they are in a frightening situation which results in mastering the skill [40].

Plausibly, humans have complex behavior, attitude, and needs, giving them diverse characteristics, which does not mean that everybody is interested in art activities [43,45]. Given art

participation across gender, studies have demonstrated a stereotype among adolescents and children that extracurricular activities like visual arts are considered girlish [43]. Evidence shows that girls build solid art self-efficacy more than boys [23,37]. Likewise, given that girls primarily participate in extracurricular activities such as painting, drawing, sculpturing, crafting, and designing that require extensive creativity. At the same time, boys choose activities like sports, music, and dance; female students establish more intensive creativity than males [14,43,45,46]. Similarly, a single study claimed no significant non-cognitive skill difference among students who did and did not participate in sports activities [47]. Nonetheless, Convey and Carbonaro's study indicates that, unlike sports activities, art participation has a notable adverse effect on non-participants [47]; an exact reflection was also observed [27].

A preceding randomized experimental study among adolescents that used panting as an intervention shown that an exceeding self-efficacy than the controlled group [39]. Indeed, individuals have a strong affection for artistic activities at a young age, given that they reflect real life [1,9,20,35]. In the same light, a recent review study claimed that Participation in art has a robust radical impact on younger children than adolescents [14,19,27,48]. Art-based activities have a role in children's identity, given that they allow them to build genuine and pleasant personalities, and visual art makes them learn and explore themselves [21]. Although several studies claimed younger subsidies from art-based activities, some exceptional studies claimed that adolescents in high schools and higher education art participation obtained promising cognitive and non-cognitive skills [28].

Artistic works involve colorful combinations that express internal emotions [1,38]. As a result, an earlier experimental study that used colors as an intervention to measure students' emotional status demonstrates that colors have a significant association with their emotions, heartbeat, and reading performance [41]. A forgoing theory states that according to the nature of artistic work, people ought to show a responsive emotional feeling [49]. Indeed, whenever people visit several artworks, it is generic to observe the genuine emotional reaction that inspires the viewers [31,42,50], although their level of cognition determines it. It has to be noted that people who tour artistic works perhaps experience negative and positive emotions depending on their cognition experience or attachment to the artwork [51]. Schaumann and his colleague's experimental study on children in Austria made them inspect different types of painting. The result showed that given that they had null awareness of abstract painting, their level of emotion was poor [35]. Additionally, individual emotion only exists as subject people are aware of reacting to it [31]. A total of 107 art museum visitors see several paintings, although hardly 4% have cognizance of the worth of the paintings [51]. Art signals people's cognitive system to direct emotional state, which ought to respond to the given artistic work [50]. Recent evidence demonstrates that children who have experience visiting or attending museums or art galleries show excellent scores in their self-efficacy [24]. This abundant evidence indicates that children's cognitive development is incomplete without art education or participation [6,8].

Furthermore, art-based interventions enhance individual creativity and social communication, which leads to positive psychological stability [2,11]. Seemingly, the creativity of individuals is the result of several psychological aspects, which are mainly centered on (cognition, emotion, and attitude) [18,43]. Children exposed to art-based intervention show a high self-esteem improvement with a shrinking negative emotional or behavioral state [11,38,41,52]. Moreover, a one-group experiment study stated that the participants surpassed confidence during the post-test, which comprised creative art activity [13]. It has to be noted that in light of the diversity of artwork, children have to participate or engage in those art-based activities which are close or applicable to their psychological capacity that results in an intended outcome [32,35]. People express different emotional statuses whenever they observe or participate

in artistic works in the house, museum, or art exhibition/gallery [19,41,52]. Notwithstanding, it has to be noted that such cognitive and non-cognitive developments can be obtained through art education is as long as children participate and that parents' socioeconomic status pointed out as the critical determiner [24,26,43,47]

Given the literature review, the study came across, which sought to test the following research hypotheses (Fig 1) in the Chinese context:

*H1*: *Girls participate in art activities out-of-school much better than boys.*

*H2*: *Student out-of-school art-based activity participation is determined by parental SES.*

*H3*: *Low SES students develop their art cognition, emotion, and self-efficacy much better than their high SES counterparts.*

*H4*: *Participating in art activities out of school directly affects art cognition and self-efficacy.*

*H5*: *Art cognition obtained from out-of-school art participation enhances students' emotions towards artistic works.*

*H6*: *Student art cognition and emotion mediate the relationship between art participation and self-efficacy.*

## Method

### Participants

The present cross-sectional survey study's data was conducted in 2022 (March–May) on junior secondary students from the rural province of Guizhou in southwestern China, consisting of four districts and two counties. The study was approved by the research ethics committee of Liaoning Normal University (Approval Number: LL2022042), and it was conducted in accordance with the approved guidelines. The target province has 106 junior secondary schools (grades 7–9), and 102 are located in rural areas. Subsequently, one Grade 9 class was sampled through the lottery method from each participating school currently living with their nuclear family, resulting in 106 participating classes. Ultimately, we ended up with a total of 2139 students ($N_{Female}$ = 1182, $N_{male}$ = 957) in grades 9 ($Mean_{Age}$ = 14.32, range = 13–17, SD = 0.561) participating in the study (see Table 1).

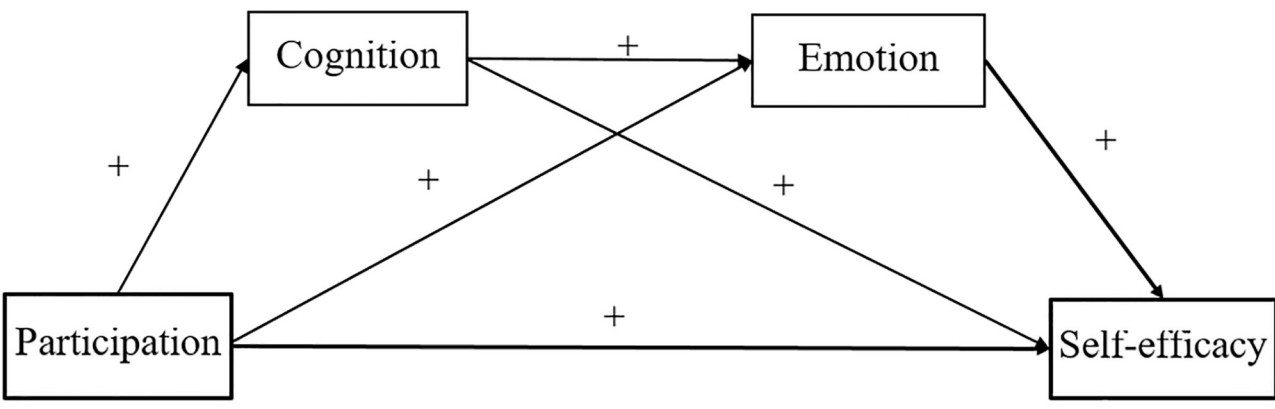

**Fig 1. Serial mediation model hypothesized for this study.**

## Ethical consideration

For data collection, proper ethical considerations have been adopted. Voluntary participation consent was obtained from the parents and legal guardians in writing before children and adolescents participated in the study. Moreover, the consent letter informs parents that all data which has been collected will be anonymous and personal information such as names. Exclusively students whose parents/ guardians authorized participation, as shown by the return of the questionnaires, were included. Table 1 shows the characteristics of the participating children. Later with the help of one author who has expertise in child psychology, explained the school teachers and administrations, the participants insisted on attempting the questionnaire translated into Chinese, which intends to measure several inquiries.

**Table 1. Participants demographic information.**

| Variables | M(SD) % |
|---|---|
| *Children Age* | 14.32 (0.561) |
| *Gender (Female)* | 1182 (55.2%) |
| *Percentage of an only child* | 14.9% |
| *Percentage of at least one parent migrant* | 53.3% |
| *Marital status* | |
| Married | 80.1% |
| Divorce | 10.5% |
| Remarried | 7.2% |
| Widowed | 2.2% |
| *Mother educational level* | |
| Illiterate | 13.1% |
| Primary School | 21.4% |
| Junior high school | 36.9% |
| Vocational High school | 6.5% |
| Senior high school | 15.2% |
| Undergraduate and above | 6.8% |
| *Father Educational Level* | |
| Illiterate | 8.6% |
| Primary School | 19.4% |
| Junior high school | 40.0% |
| Vocational High school | 7.4% |
| Senior high school | 16.4% |
| Undergraduate and above | 8.2% |
| *Mother monthly income* | |
| Under 1000RMB | 28.8% |
| 1000-3000RMB | 44.0% |
| 3000-6000RMB | 20.6% |
| 6000-9000RMB | 4.4% |
| Over 10000RMB | 2.2% |
| *Father Monthly Income* | |
| Under 1000RMB | 16.1% |
| 1000-3000RMB | 40.4% |
| 3000-6000RMB | 28.5% |
| 6000-9000RMB | 11.0% |
| Over 10000RMB | 4.0% |

## Measurements

The participants' socioeconomic status was measured with two common indicators: parents' educational levels and the total parents' monthly income. A large number of studies measure parental educational levels (father and mother) with 4- to 5-point Likert scales. However, since the present study discusses the rural context, the range of parental educational attainment may be more comprehensive. Accordingly, parental educational level (for both fathers and mothers) was captured with a 6-point Likert scale: 1 = *No education (illiterate)*, 2 = *primary school*, 3 = *Junior High school*, 4 = *Technical secondary school/vocational high school*, 5 = *Senior high school*, 6 = *undergraduate, and above*. The internal consistency reliability coefficients (Cronbach's alpha) were 0.81, whereas 0.79 for the fathers and 0.83 for the mothers. The distribution of parental educational levels across parents confirms the study speculation that rural Chinese parents' educational levels are not normally distributed. The other SES indicator is monthly parental income, which was recorded from responses by each participating student's parents or legal guardians (grandparents) on a five-point Likert scale: 1 = *under 1000 RMB*, 2 = *1000– 3000 RMB*, 3 = *3000–6000 RMB*, 4 = *6000–9000 RMB*, 5 = *over 9000 RMB* (See Table 1).

**Art cognition.** The authors designed a questionnaire of five-item measured by four scales Likert scales ranging from 1 = *very unfamiliar* to 4 = *very familiar*; these five items are intended to measure the extent to which participants know art. Some of the sample items from the survey are: "I know how to highlight the subject in painting.", "I have a full clue about different visual art categories," "I am good at drawing with many visual elements such as lines, shapes, spaces, and colors." The value of Cronbach's alpha of this parameter was 0.76.

**Art emotion.** Similar to cognition, in this study questionnaire, the authors designed and validated items that measure students' affection or emotional attachment to artistic work by five items with four scales Likert scale ranging from 1 = *very unfamiliar* to 4 = *very familiar* with Cronbach's alpha value of 0.801. Some of the sample items are: "I like to make some small works of art.", "I will express the beauty artistically found in my life.", "I often have an emotional resonance with art creators."

**Art participation.** To assess participants' likelihood of participating in diverse art-based activities, five items were comprised that were measured by four Likert scales ranging from 1 = *never participate* to 4 = *regularly participate*. A few of the sample items are: "I will express the beauty artistically found in my life.", "I often go to art exhibitions or community cultural activities.", "I attend all kinds of courses related to art.", "I like to photograph natural landscapes, buildings, and figures with aesthetic and artistic feeling." The value of internal consistency reliability coefficients was 0.74.

**Self-efficacy.** To measure participants' art self-efficacy, the study adopted the most classic and reliable instrument developed in Germany for almost three decades (Schwarzer & Jerusalem, 1995). The survey consists of 10 items with a 5 Likert scale ranging from 1 = *very inconsistent* to 5 = *very consistent*, with sample items: "Even if others oppose me, I still have a way to get what I want.", "I can face difficulties calmly because I can rely on my ability to deal with problems.", "No matter what happens to me, I can cope with it." In the present study, a 0.77 Cronbach alphas value was obtained.

## Data analysis procedures

First, a two-way Multivariate Analysis of Variance (MANOVA) was conducted to examine the association among the study variables. Second, Pearson correlation analyses were conducted to investigate the associations among art participation, cognition, art emotion, and self-efficacy. Third, AMOS 21.0 was utilized to test the serial mediation model. The bootstrap method based on 2000 samples was used to obtain bias-corrected and accelerated 95% confidence

**Table 2. Confirmatory factor analysis results.**

| Variables | $X^2$ | Df | CFI | GFI | RMSEA | SRMR |
|---|---|---|---|---|---|---|
| Art Participation | 863.18 | 2125 | 0.923 | 0.976 | 0.039 | 0.0113 |
| Art Cognition | 890.27 | 2119 | 0.965 | 0.988 | 0.041 | 0.0257 |
| Art Emotion | 916.62 | 2120 | 0.917 | 0.992 | 0.047 | 0.0190 |
| Art Self-efficacy | 821.3 | 2123 | 0.942 | 0.987 | 0.029 | 0.046 |

**GFI**, goodness-of-fit index; **CFI**, comparative fit index; **RMSEA**, root-mean-square error of approximation; **SRMR**: Standardized Root Mean Square Residual.

intervals (CI) for the serial mediation model. In addition, a series of confirmatory factor analyses were conducted to test the unidimensionality of the latent factors, and three alternative measurement models were compared with the hypothesized baseline model. The results, as presented in Table 2, indicated that the hypothesized four-factor measurement model fit the data better than the alternative models, with $\chi2$ = 810.240, df = 2123, p < 0.001, CFI = 0.974, GFI = 0.979, RMSEA = 0.029, and SRMR = 0.046.

## Results

### Descriptive statistics

The Means and standard deviations of the variables are presented in Table 3. In addition, a two-way MANOVA was conducted to examine the association between children's profile and their closeness to art education in Table 4. The two-way MANOVA result shed light on the fact that there is a significant art participation difference between boys and girls ($F$ (1, 2139) = 21.32, $p$ < .05), which confirms our first speculation. In addition, regarding household characteristics, the father's educational level has an adverse effect on students' cognition and emotion they possess towards Art ($F$ (1, 2139) = 30.01, $p$ < .01) and ($F$ (2, 2139) = 14.20, $p$ < .05) respectively. Given mothers' educational level, the MANOVA result demonstrates an unexpected finding that mothers' level of academic study cannot predict their children's likelihood of art activity participation compared to previous Chinese literature ($F$ (1, 2139) = 3.01, $p$ = .090), which partially negates the second hypothesis. Moreover, like the father's education rank, the mother's education also has a significant association with children's art cognition ($F$ (7, 2139) = 7.15, $p$ < .001) and emotion ($F$ (8, 2139) = 5.012, $p$ < .05), which support the third assumption of this study. Moreover, a significant variance of art participation difference was observed among boys and girls ($F$ (1, 2139) = 5.192, $p$ < .05) but not between Han and minority races ($F$ (7, 2139) = 4.031, $p$ = 0.079).

**Table 3. Descriptive result.**

| | Total | Male | Female |
|---|---|---|---|
| Sample | 2139 | 957 (44.7%) | 1182 (55.3%) |
| SE | 3.22 (0.751) | 3.31 (0.769) | 3.14 (0.727) |
| AP | 1.28 (0.510) | 1.32 (0.562) | 1.25 (0.462) |
| AE | 2.47 (0.649) | 2.37 (0.697) | 2.55 (0.596) |
| AC | 2.24 (0.555) | 2.18 (0.605) | 2.28 (0.507) |

*Note*. SE: Self-efficacy, AP: Art Participation, AE: Art Emotion, and AC: Art cognition.

**Table 4. Two-way MANOVA result.**

| | Art Cognition | | Art Participation | | Art Emotion | | Self-Efficacy | |
|---|---|---|---|---|---|---|---|---|
| | *F* | *SE* | *F* | *SE* | *F* | *SE* | *F* | *SE* |
| Gender_2Female | 21.32* | 0.587 | 5.192* | 0.980 | 2.061 | 0.442 | 2.901 | 7.424 |
| ME | 7.15*** | 0.708 | 3.01 | 8.821 | 8.021* | 0.806 | 1.048 | 1.042 |
| FE | 30.01** | 0.146 | 3.52 | 3.515 | 7.20* | 0.629 | 2.991 | 0.201 |
| HI | 3.986 | 6.275 | 9.36 | 7.413 | 3.03 | 1.414 | 1.05 | 9.358 |
| Race_2minority | 5.05*** | 0.784 | 4.03 | 0.698 | 6.05* | 0.904 | 2.021 | 7.314 |

*Note*:

*$p < .05$,

**$P < 0.01$,

***$p < .001$.

ME = Mother Education level; FE: Father Education level; HI = Monthly household income; Gender code = 1: Male, 2: Female; Race Code = 1: Han race, 2: Minority.

Subsequently, a bivariate Pearson correlation was run among art cognition, participation, emotion, and self-efficacy, discussed in Table 5. The result shows that children's art participation has a significant relationship with their growth of art self-efficacy ($r = 0.507$, $p < .001$), art emotion ($r = 0.594$, $p < .001$), and art cognition ($r = 0.376$, $p < .001$), which validates their fourth hypothesis of this study. Likewise, the bivariate analysis indicates that their cognition significantly influences children's art self-efficacy ($r = 0.466$, $p < .001$) and emotion toward Art ($r = 0.531$, $p < .001$).

## Confirmatory factor analysis

Confirmatory factor analysis (CFA) was performed for each variable (Art Participation with five items, Art cognition with five items, art emotion with five items, and self-efficacy with ten items). All factor loading values are believed to be greater than 0.5, indicating that each variable fits well (Kline, 2005). The results of CFA showed that all factor loading values range from 0.603 to 0.897, greater than 0.5. Specifically, art emotion showed a good model fit (Chi-square/df = 0.432, SRMR = 0.0190, GFI = 0.992, CFI = 0.917). So is art cognition (Chi-square/df = 0.420, SRMR = 0.0257, GFI = 0.988, CFI = 0.965). Art participation also meets the requirement with Chi-square/df = 0.406, SRMR = 0.0113, GFI = 0.976, CFI = 0.923, together with self-efficacy (Chi-square/df = 0.386, SRMR = 0.046, GFI = 0.987, CFI = 0.942) (See Table 2).

## The effect of art participation on the development of self-efficacy

The present study adopted the path analysis or structural equation model to assess children's direct and indirect effects on their self-efficacy with the multiple mediations of art cognition

**Table 5. Bivariate correlation among variables.**

| | *M* | *SD* | 1 | 2 | 3 | 4 |
|---|---|---|---|---|---|---|
| SE | 3.22 | 0.751 | 1 | | | |
| AP | 1.28 | 0.510 | 0.507*** | 1 | | |
| AE | 2.47 | 0.649 | 0.466*** | 0.594*** | 1 | |
| AC | 2.24 | 0.555 | 0.531*** | 0.376*** | 0.396*** | 1 |

Note:

***p < .001.

SE: Self-efficacy, AP: Art Participation, AE: Art Emotion, and AC: Art cognition.

and emotion. Therefore, a structural equation model (SEM) was performed with AMOS 21.0 on a dataset with no missing values. The results from the SEM indicate a good model fit; the chi-squared ($\chi^2$) value was highly significant ($p < .001$), as were the values of the goodness-of-fit index (GFI; 0.94), the comparative fit index (CFI; 0.92), and the mean square error of approximation (RMSEA; 0.012), which implies that the model fits the data well. The present study's SEM results indicate that the direct effect of children's art participation on their self-efficacy is significant ($\beta = 0.06$, $p < .01$) and accounted for 46.1% of the variance. Further, the path analysis result unfolds that art participation can significantly predict children's art cognition ($\beta = 0.376$, $p < .001$) and emotion ($\beta = 0.052$, $p < .05$). Similarly, art emotion ($\beta = 0.133$, $p < .001$) and cognition ($\beta = 0.056$, $p < .05$) have a considerable influence on children's growth of self-efficacy, which is in line with the speculation stated in the fifth assumption of the study.

### The mediating role of art emotion and cognition on the relationship between art participation and self-efficacy

The bootstrapping method tested to test the mediating effect of art cognition and emotion on the connection between art participation and self-efficacy. This procedure encompasses eight models that estimate 95% confidence intervals for these effects from 2000 resamples of the data, and those confidence interval values that do not include zero within the upper and lower bounds imply a significant $p$-value less than 0.01. Figs 1 and 2, and Table 6 illustrate that the indirect effect size of art participation on self-efficacy in the light of cognition and emotion mediation is ($\beta = 0.019$, 95% CI ranging from 0.021 to 0.042). In addition, the result revealed that art participation has a significant indirect effect on self-efficacy with the mediating role of art cognition ($\beta = 0.021$, 95% CI ranged from 0.012 to 0.030). The indirect effect of participation → Emotion → Self-efficacy is also significant ($\beta = 0.007$, 95% CI ranging from 0.071 to 0.092). This finding notes that in the serial mediation model, children's art cognition has a more vital mediating role than emotion in the association between art participation and self-efficacy, which support the six hypotheses of the present study.

## Discussion

### Art participation and socioeconomic status

Surprisingly, the multivariate (MANOVA) analysis demonstrates that boys participate more in out-of-school art than girls in junior secondary schools in the Chinese context. This finding

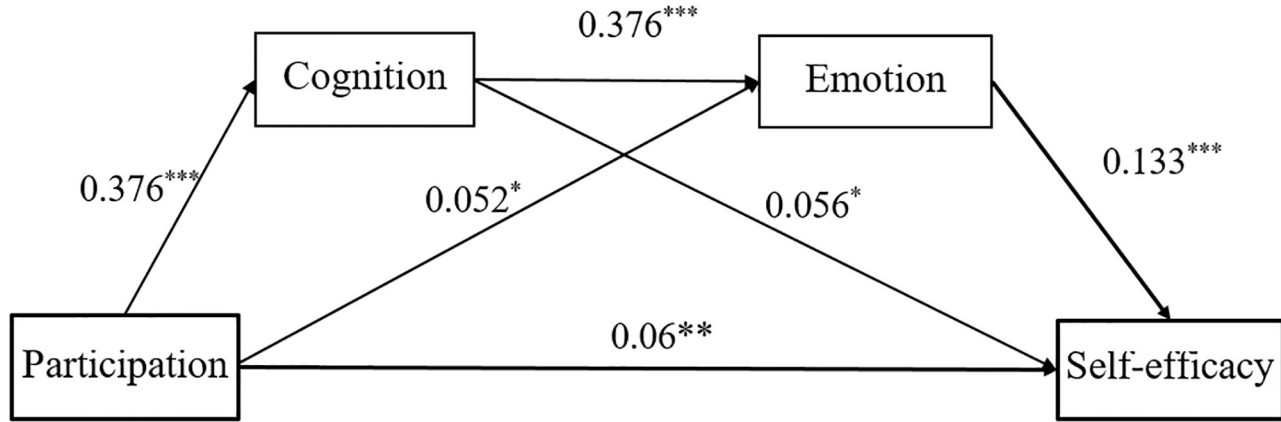

**Fig 2. The mediating role of art cognition and emotion on the relationship between art participation and self-efficacy.** Note: *p < .05, **p < .01, ***p < .001.

**Table 6. The direct and indirect effects of art participation on self-efficacy.**

| Paths | Effect | 95% CI |
|---|---|---|
| Participation → Self-Efficacy | 0.060** | [0.014, 0.0041] |
| Participation → Cognition → Self-Efficacy | 0.021** | [0.012, 0.030] |
| Participation → Emotion → Self-Efficacy | 0.007** | [0.071, 0.092] |
| Participation → Cognition → Emotion → Self-Efficacy | 0.019*** | [0.024, 0.027] |

Note:
*$p < .05$,
**$p < .01$,
***$p < .001$.

negates a large body of international evidence that states that art-based activities are highly preferred and stereotyped by girls from extracurricular activities children are involved in after school [18,19,23,46]. The possible explanation for this finding is that, unlike in Western countries, in most Asian countries, parents favor boys or son over the daughter due to the cultural preference for boys to provide the best facility for their future academic and professional advantage that brings the high rate of return for the parents at the old age. Unfortunately, the two-way MANOVA result indicates that boys have not shown any significant difference in art cognition, emotion, and self-efficacy with their girl counterparts. This exciting finding might tell us that boys attend out-of-school art schools without interest; studies noted that children engaged in extracurricular activities due to strict parental command with no child interest degrade art education merit.

Astonishingly, this study contradicts a large volume of international claims that high SES children attend out-of-school art activities, which require a hefty fee and intensive support from parents [2,28,39,43]. Given the parental SES (educational level and income) of junior secondary students' participants, the speculation for this finding is that over three-fourths of parents possess only primary and junior secondary education, which explains their monthly income. Although a prior Chinese study stated that a mother's education and income have a stronger association with children cultural capital than a father, the present study couldn't support the argument that the target region of this study's SES was moderate. Therefore, this study may not show that socioeconomic status influences out-of-school art activity participation due to no significant SES difference among the society. Likewise, the present study has not shown any substantial evidence that stresses the literature's argument that disadvantaged children can benefit from art-based activities more than high SES.

Therefore, the first three hypotheses in this section were tested, and the findings failed to support them. Instead, the present study disputes a considerable amount of international literature, which shows that visual art and social culture are highly interrelated.

## The mediating role of art cognition and emotion on the relationship between art participation and self-efficacy

The structural Equation Model (SEM), or path analysis, investigates the direct and indirect effect of out-of-school art participation on children's self-efficacy by mediating cognition and emotion. Primarily, the SEM analysis finding confirms the hypothesis and previous literature that Participation in art promotes individual cognition, emotion, and efficacy [17,42,43,44]. In addition, the finding revealed that individual cognition has a positive influence on their emotional state, which supports previous evidence and the hypothesis, which states that unless

individuals have a solid art cognition through stable art participation, there is no way they can exhibit substantial emotional affection towards artistic work [2,31,34]. Consequently, the study showed such positive relationships between art cognition and emotion that led them to obtain a decent self-efficacy built through continued art participation which promotes the knowledge they have of art and expresses the emotion or feeling whenever they sense (see tough, listen) [4,13,38].

Because cognition and emotion mediate the association between art participation and self-efficacy, the SEM model result supports the hypothesis that cognition and emotion levels determine the strength of the relationship between art participation and self-efficacy. Mainly, art cognition's mediating role is more potent than emotions. Similarly, emerging evidence states that whenever children engage in artistic activities, they start to develop a habit of communicating with their peers, showing their work, comment works that are made by their peers or observed in art exhibitions or museums; such process makes them self-efficacious [15,21,37,39]. Ultimately, this paper finds that besides academic benefits, art education promotes nonacademic skills, which are the primary agents for academic performance [6,20,46,47].

## Conclusion

The present study sheds light on Chinese junior secondary students' likelihood of art-based participation outside the school at privately owned schools. The paramount aim of the study was: to show the association between out-of-school art participation and self-efficacy with the mediation of cognition and emotion. Based on the rigor analysis and findings, the study concludes that the participation of children and adolescents in visual art activities fosters their cognition about artistic knowledge, emotional affiliation, and affection with several aesthetic works, and most significantly, it boosts the self-belief and esteem inform of others on themselves which Chinese adolescents lack. Although this study unfolds the potential impact of visual art activities on adolescents' well-being, it must be noted that only financially capable parents can send their children to such out-of-school visual art centers.

## Implications, limitations, and future research directions

Herein according to the finding, robust policy and practical implications are forwarded. The present study's first practical implication is that Chinese parents' ought to value boys and girls with the same eye regarding providing any extracurricular activities outside of school. Thus, the government and stallholders have to spread awareness concerning the application of art education in society to create a mindset that appreciates children's art-based activities regardless of their sex, given future hope of the country depends on today's children (boys and girls) psychological, physical and social well-being. In the same light, while parents start changing sex-role perception by involving their children in extracurricular activities that they are interested in, the government has to foster affordable or accessible extracurricular institutions or places out-of-school that children can come to after school and boost their artistic or any skill they have without thinking of financial constrain. In addition, art-based activities can be used in the school and other institutes to enhance children's self-esteem, peer-relationship, positive emotion, and self-esteem. Since Chinese children are too attached to electronic devices than their peers, art can be a reliable intervention to develop children's social skills even in school. Moreover, the study suggests a policy implication: the government can promote an in-school art curriculum instructed by qualified teachers to restore parity among students who can and cannot afford to participate.

Although the study used a large-scale survey from junior secondary schools in Guizhou province, located in the Southwestern part of China, there is a couple of limitation the study could not address, which is recommended for scholars as a future study direction. The first limitation is that the study target only one province in the southwestern part of China, which leaves us with 25 other provinces in China where the findings cannot be generalized. So future studies are needed in other provinces to underline the extensive application of art education. In addition, given that this paper conducted a one-time survey to examine the relationship among the variables, it is proscribed from claiming casual relationships. The current study suggests future studies in China that target extracurricular activities and gender, children's interests, parental support, and socioeconomic status. Third, due to the nature of the research design, the study could not provide in-depth evidence regarding the perception and experience of participants in taking part in visual art and other extracurricular activities, which can be obtained through exploratory study.

## Supporting information

**S1 Data.**
(SAV)

## Acknowledgments

We want to express our great appreciation and admiration for the adorable children who were part of this study and their parents and teachers who made the data collection happen.

## Author Contributions

**Conceptualization:** Genman Deer, Endale Tadesse, Sabika Khalid, Chunhai Gao.

**Formal analysis:** Genman Deer, Zhihan Chen, Chunhai Gao.

**Investigation:** Endale Tadesse, Zhihan Chen, Chunhai Gao.

**Methodology:** Genman Deer, Endale Tadesse, Zhihan Chen, Chunhai Gao.

**Resources:** Sabika Khalid.

**Software:** Sabika Khalid.

**Supervision:** Genman Deer, Chunhai Gao.

**Writing – original draft:** Endale Tadesse, Sabika Khalid, Chunhai Gao.

**Writing – review & editing:** Genman Deer, Chunhai Gao.

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
