## [Decision Letter · Decision Letter 0]

26 May 2023

PONE-D-23-06961The Impact of Chinese Adolescents Visual Art Participation on Self-Efficacy: A Serial Mediating Role of Cognition and EmotionPLOS ONE

Dear Dr. Tadesse,

Thank you for submitting your manuscript to PLOS ONE. After careful consideration, we feel that it has merit but does not fully meet PLOS ONE’s publication criteria as it currently stands. Therefore, we invite you to submit a revised version of the manuscript that addresses the points raised during the review process.

We look forward to receiving your revised manuscript.

Kind regards,

Yuh-Yuh Li, Ph.D.

Academic Editor

PLOS ONE

Journal Requirements:

Reviewers' comments:

Reviewer's Responses to Questions

**Comments to the Author**

1. Is the manuscript technically sound, and do the data support the conclusions?

Reviewer #1: Yes

Reviewer #2: Yes

Reviewer #3: Partly

Reviewer #4: Partly

2. Has the statistical analysis been performed appropriately and rigorously? 

Reviewer #1: Yes

Reviewer #2: Yes

Reviewer #3: No

Reviewer #4: No

3. Have the authors made all data underlying the findings in their manuscript fully available?

Reviewer #1: Yes

Reviewer #2: Yes

Reviewer #3: Yes

Reviewer #4: Yes

4. Is the manuscript presented in an intelligible fashion and written in standard English?

Reviewer #1: Yes

Reviewer #2: No

Reviewer #3: No

Reviewer #4: No

5. Review Comments to the Author

Reviewer #1: The Research technically sound, & the data support the conclusions

The Statistical analysis performed appropriately and rigorously

The Authors made all data underlying the findings in their manuscript fully available

The Manuscript presented in an intelligible fashion and written in standard English, The way of writing very clear, correct, & unambiguous

Reviewer #2: Abstract

The abstract reads well and captures the essence of the study.

Introduction:

This section reads more like a review of literature than an introduction. There were far too many references to existing literature, indicating that this study was based on a thorough review of related literature. It is expected that the authors had a reason for conducting a literature review in the first place. That is what we are expecting to see in the introduction. That will be the impetus for the study. This could be conceptualised better without relying on too many references to related literature.

It is worth noting that, contrary to the authors' assertion, some extracurricular activities can be relatively inexpensive. That could be why children from low-income families excel in professional football, a sport that is considered extracurricular at the primary school level.

It might be useful to mention the range of skills that a child is likely to acquire at primary/secondary school age, as well as how art can stimulate those skills.

I noticed multiple sources (citations) for each claim, indicating thorough research. However, you might want to limit it to three or four references. In-text citations abound on the first few pages.

Literature Review:

The array of references from related literature demonstrates the breadth of the research here. However, the literature review should be more critical. It is dense with references to other people's work, with little input from the authors on whether they agree or disagree with the positions in the existing literature. The references (works cited) must be contextualised within the Chinese environment - why do the authors agree with those positions? And why don't they agree based on local observation? Again, a portion of the literature review should have been included in the introduction section, and similar arguments were presented in both the introduction and the literature review section.

Secondly, the literature review section should be organised in the same order that the concepts were introduced: art participation, cognition, emotion, and self-efficacy. As a result, I would have expected a conceptual approach in the section on literature review.

There were a few spelling mistakes, sentence construction issues, and grammatical errors in this and the other sections. On the fourth page of the literature review section, there were a couple of incomplete sentences, such as "Likewise, given that primarily girls participate in extracurricular activities such as painting, drawing, sculpturing, crafting, and designing that require extensive creativity."

Hypotheses:

This is concise and clear. The structure of the literature review, on the other hand, should have followed the same order as the hypothesis presented. This will facilitate reading and improve comprehension.

Materials and Methods:

This section is well-written and lends credibility to the researchers' expertise. But what about the research design? And why do you call this research "cross-sectional"?

Findings:

A logical approach to presenting the findings in such a way that they align with the hypothesis would have facilitated reading and comprehension. To address this, I propose that the findings be restructured.

Discussion

I noticed that the six hypotheses were divided into two broad categories in the discussion section. While this is good, it may also be beneficial to inform the readers of the reasoning behind this decision.

Concerning the statement, "Ultimately, this paper discovers that, in addition to academic benefits, art education promotes nonacademic skills, which are the primary agents for academic performance," I find it difficult to associate nonacademic skills as an agent of academic performance. This statement should be clarified in the final paragraph of the discussion section.

Conclusion:

While the conclusion discusses the relationship between art cognition and emotion, it is silent on the relationship between participants' socioeconomic status and art participation. The conclusion should address the two broad areas of focus of this study in order to be complete.

On the last page of your ethical statement, double-check the spelling. The ethical statement, not the ethnic statement, should be used.

Reviewer #3: 1) This paper examines “the potential impact of visual art on adolescents’ emotional, cognition and self-efficacy development” and “the association between parental SES and children’s likelihood to participate in art education and its impact on children’s psychological well-being Emotion, Cognition, and Self-efficacy)

2) SES was measured by each mother’s and father’s educational level and parent’s monthly income. The authors report internal consistency reliability coefficients for these measures but do not indicate why these are needed or their usefulness.

3)Then, the authors define three constructs: art cognition, art emotion, and art participation. For each construct, a 5-item questionnaire measured on a 4-point Likert scale was designed.

Lastly, the authors measured Self-efficacy by adopting “the most classic and reliable instrument developed in Germany for almost three decades (Schwarzer & Jerusalem, 1995).” Unfortunately, the authors do not indicate the name of this test, nor do they present the psychometric properties of it. Obviously, this self-efficacy test was translated to the dialect of the region where the data were collected, but the authors did not provide any information about this process or present any validation data to show the validity of the translated test.

4) The authors did not provide sufficient psychometric data and analysis to show the extent to which these four constructs (art cognition, art emotion, art participation, and self-efficacy) are independent of one another. There is no data to support the reliability and validity of these measures. The dataset is sufficiently large to conduct factor analysis to check the independence and utility of these constructs and provide some psychometric properties and statistical validation.

5) I shall refrain from reviewing the MANOVA results until I know more about the abovementioned four constructs.

6) The authors seem to argue that art cognition, art emotion, and art participation are related to cognitive and emotional development and, ultimately, self-efficacy. However, these connections are not clearly made and focused upon in the introduction or the discussion. The test jumps between ideas without a clear focus.

7) The In-text citation is inconsistent APA (alphabetical order of references and the use et al with more than three authors)

8) This paper reads more like a mini-thesis, not an article intended for publication in a journal.

9) The literature review is long and unnecessary.

10) The ideas presented in the introduction are repeated in the literature review. The introduction and the literature review should be combined into one section, “Introduction,” which should end with the rationale and justification for the study. The current introduction and literature review should be summarized into four to five pages at most.

11) The authors often use the phrase “shred of studies” to refer to few or scattered studies. Perhaps, using a more commonly used phraseology is encouraged.

12) The Participants section includes the following statement: “… 2139 students (NFemale = 1182) in grades 9 (MeanAge = 14.32, range = 13-17, SD = 0.561) participating…” Why report the female but not the male participants?

13) Why is it “exciting” that mother’s educational level cannot predict art participation?

14) This study has more limitations than restricting the data collection to one province and one survey. The limitation

Many grammatical errors

15) In summary, this manuscript needs to be rewritten as a journal article with a clear and precise focus on the objectives of the research, the relationship between the objectives and previous research findings, the psychometric properties of the four constructs, and the relationship between the findings of this study and previous research.

Reviewer #4: Please find below my observations and recommendations.

1. Refer to the PLOS One journal submission guidelines for referencing and other journal reporting formats.

2. It appears the introduction and literature review are not in order. Some ideas are repeated several times in different paragraphs. e.g., art education's influence on academic and psychological development It could be refined and presented in a concise and orderly manner (besides, this is not a systematic review). Again, most discussions are unilateral, ignoring other insights that would properly inform readers; e.g., even though art education or activities improve academics, there are several areas of academics where there are no reliable causal links, and the same is true for other literature discussed. I recommend that conciseness and coherency be prioritised, and I also recommend taking away old studies (citations) where there is much current literature to support a claim.

3. Additional professional editing is needed to improve the manuscript.

4. A reliability test alone is not enough to fully validate a new instrument (or a construct). Other necessary validation analyses, such as EFA and CFA, may be needed to address the various aspects of validity.

5. The language medium of the instruments is unknown, as is the duration of the data collection. It would be more appreciated if each section in the methodology addressed specific issue(s). For example, issues regarding ethical consideration are scattered. Ethical procedures should also be named appropriately.

6. Your strict inference that Chinese parents are definitely responsible for what is observed should be considered (under implications, second line). It could be due to several factors.

6. PLOS authors have the option to publish the peer review history of their article (what does this mean?). If published, this will include your full peer review and any attached files.

Reviewer #1: **Yes: **Dr.Ali Hamood Twaij

Reviewer #2: No

Reviewer #3: **Yes: **Dr. Ali M. AL-Asadi

Reviewer #4: No

---

## [Author Response · Author response to Decision Letter 0]

30 May 2023

Response to Reviewers' Comments

Dear Editors and reviewers,

 I am writing this letter on behalf of my coauthor regarding our manuscript PONE-D-23-06961, entitled " The Impact of Chinese Adolescents Visual Art Participation on Self-Efficacy: A Serial Mediating Role of Cognition and Emotion." We want to express our appreciation to the respected editors and reviewers for providing us the constructive comments and suggestions to shape our manuscript for quality publication. According to our original reviewer's last review, we have obtained critical and rational minor suggestions to complete the manuscript's structure and content before the official publication. As you can see below, we have responded to each comment given by the reviewer corresponding to each page's revision has been made.

Reviewer 1 Comments

Comment 1: The Research technically sound, & the data support the conclusions

The Statistical analysis performed appropriately and rigorously

The Authors made all data underlying the findings in their manuscript fully available

The Manuscript presented in an intelligible fashion and written in standard English, The way of writing very clear, correct, & unambiguous

Response: We genuinely appreciate the profound admiration for the work, which significantly benefits us.

 Reviewer 2 Comments

Comment 1: The abstract reads well and captures the essence of the study.

Response: Thank you for the appreciation.

Comment 2: There were far too many references to existing literature, indicating that this study was based on a thorough review of related literature. It is expected that the authors had a reason for conducting a literature review in the first place.

Response: Thank you for the reminder; we made sure that the number of citations is managed to keep the concept flow.

 Comment 3: It is worth noting that, contrary to the authors' assertion, some extracurricular activities can be relatively inexpensive.

Response: We considered the suggestion. Unlike inexpensive extracurricular activities, the literature addressed the role of visual art in developing children's cognitive and non-cognitive development.

Comment 4: the literature review section should be organised in the same order that the concepts were introduced: art participation, cognition, emotion, and self-efficacy. As a result, I would have expected a conceptual approach in the section on literature review

Response: Per the suggestion of the respected reviewer, we managed to organize the literature review based on the structure of the aim of the study.

Comment 5: There were a few spelling mistakes, sentence construction issues, and grammatical errors in this and the other sections.

Response: We would like to note that a professional editing service has been used to restore the language accuracy and quality of the manuscript. 

Comment 6: And why do you call this research "cross-sectional"?

Response: Since the study conducted a one-time survey. The study design is a cross-sectional survey design.

Comment 7: A logical approach to presenting the findings in such a way that they align with the hypothesis would have facilitated reading and comprehension. To address this, I propose that the findings be restructured.

Response: We considered the suggestion and the study's finding constructed based on the order of the hypotheses.

Comment 8: I find it difficult to associate nonacademic skills as an agent of academic performance. This statement should be clarified in the final paragraph of the discussion section.

Response: We intended to state that most of the study investigated the application of visual art on children's academic performance, but our study showed evidence of non-academic outcomes.

 Comment 9: The conclusion should address the two broad areas of focus of this study in order to be complete.

Response: Thank you for the suggestion. The conclusion is revised. 

Reviewer 3 Comments

Comment 1: The authors report internal consistency reliability coefficients for these measures but do not indicate why these are needed or their usefulness.

Response: Thank you for the suggestions and comments; we learned that it was unnecessary.

Comment 2: The dataset is sufficiently large to conduct factor analysis to check the independence and utility of these constructs and provide some psychometric properties and statistical validation.

Response: We appreciate the reviewer for pointing out the essential suggestion. Kindly refer to Table 2 and the finding section for the factor analysis to test whether the data fit a hypothesized measurement model.

Comment 3: This paper reads more like a mini-thesis, not an article intended for publication in a journal.

Response: We conducted an intensive organization of literature review per the request of the reviewers’ suggestion. 

Comment 4: The Participants section includes the following statement: “… 2139 students (NFemale = 1182) in grades 9 (MeanAge = 14.32, range = 13-17, SD = 0.561) participating…” Why report the female but not the male participants?

Response: As long as the total number of participants is mentioned, it is obvious to know that participants after the female are male participants.

Comment 5: Why is it “exciting” that mother’s educational level cannot predict art participation?

Response: We have mentioned in the literature and discussion section that in the Chinese context Mother's education level has a strong determiner of the child's quality of life compared to the father's economic status.

Reviewer 4 Comments

Comment 1: Refer to the PLOS One journal submission guidelines for referencing and other journal reporting formats.

Response: Suggestion has been taken and we used the journal reference style.

Comment 2: It appears the introduction and literature review are not in order. Some ideas are repeated several times in different paragraphs

Response: The suggestion has been considered; we rearranged and organized the flow of the manuscript. 

 . 

Comment 3: Additional professional editing is needed to improve the manuscript

Response: Suggestion has been taken. A professional editing service is used.

 Comment 4: A reliability test alone is not enough to fully validate a new instrument (or a construct). Other necessary validation analyses, such as EFA and CFA, may be needed to address the various aspects of validity.

Response: Suggestions have been considered. Kindly check Table 2 and the CFA analysis report in the result section.

Comment 5: The language medium of the instruments is unknown, as is the duration of the data collection. It would be more appreciated if each section in the methodology addressed specific issue(s). For example, issues regarding ethical consideration are scattered. Ethical procedures should also be named appropriately.

Response: We addressed such important information per the reviewer's suggestion.

Comment 6: Your strict inference that Chinese parents are responsible for what is observed should be considered (under implications, second line). It could be due to several factors.

Response: The suggestion is considered and addressed.

---

## [Decision Letter · Decision Letter 1]

19 Jun 2023

PONE-D-23-06961R1The Impact of Chinese Adolescents Visual Art Participation on Self-Efficacy: A Serial Mediating Role of Cognition and EmotionPLOS ONE

Dear Dr. Gao,

Thank you for submitting your manuscript to PLOS ONE. After careful consideration, we feel that it has merit but does not fully meet PLOS ONE’s publication criteria as it currently stands. Therefore, we invite you to submit a revised version of the manuscript that addresses the points raised during the review process. 

We look forward to receiving your revised manuscript.

Kind regards,

Yuh-Yuh Li, Ph.D.

Academic Editor

PLOS ONE

Journal Requirements:

Reviewers' comments:

Reviewer's Responses to Questions

**Comments to the Author**

1. If the authors have adequately addressed your comments raised in a previous round of review and you feel that this manuscript is now acceptable for publication, you may indicate that here to bypass the “Comments to the Author” section, enter your conflict of interest statement in the “Confidential to Editor” section, and submit your "Accept" recommendation.

Reviewer #2: All comments have been addressed

Reviewer #3: (No Response)

Reviewer #4: (No Response)

2. Is the manuscript technically sound, and do the data support the conclusions?

Reviewer #2: Yes

Reviewer #3: Yes

Reviewer #4: Partly

3. Has the statistical analysis been performed appropriately and rigorously? 

Reviewer #2: I Don't Know

Reviewer #3: Yes

Reviewer #4: No

4. Have the authors made all data underlying the findings in their manuscript fully available?

Reviewer #2: Yes

Reviewer #3: Yes

Reviewer #4: Yes

5. Is the manuscript presented in an intelligible fashion and written in standard English?

Reviewer #2: Yes

Reviewer #3: Yes

Reviewer #4: No

6. Review Comments to the Author

Reviewer #2: The Authors have made efforts in addressing all the concerns raised during the first round of review.

Reviewer #3: 1) It is understood and expected that a literature review is part of the introduction. The heading “Literature review” should be removed, and a proper transition that leads to the heading “Art participation…” should be inserted.

2) Again, stating the number of males and females in the sample is customary and appropriate. Stating one but not the other expects the reader to figure it out! Please state the number of both sexes.

3) When data analysis leads to “exciting” results, the reasons for such excitement should be explicitly stated. The authors highlighted the following sentences to explain why the results are exciting: “which partially negates the second hypothesis,” which supports the third hypothesis,” and “which validate their (it should be the) fourth hypothesis of this study.” I do not believe this is informative, and again, it burdens the reader to figure out why the authors think these results are wonderful. We should always keep the readers in mind and write for them.

4) While Table 2 provides the confirmatory factor analysis results, the very few words at the end of the Data analysis procedures are hardly sufficient to address the validity of these scales. Moreover, only the fact that confirmatory factor analysis is performed to validate the hypothesized scales should be stated under the data analysis procedure. The factor analysis results should be part of the results section under its own subheading after descriptive statistics. This part should be thoroughly presented in relation to the validation of the scales. That is, explain the results found in Table 2 and reference them to Table 2.

5) I do not believe stating chi-squared to have a highly significant value without properly stating the parameter’s value is consistent with PLOS style.

Reviewer #4: 1.“Hence, discussing out-of-school extracurricular activities' influence on children's cognitive and non-cognitive development took the attention of scholars the present study sought to investigate.” Please consider this sentence. It seems not to be clear. Consider ensuring the free flow of understanding in that paragraph.

1b, "Furthermore, it must be stressed that besides the cognitive development of children through art education, also it strengthens the non-cognitive abilities (self-esteem, self-concept, self-efficacy,

peer-relationship, emotions)," (Please replace "also it" with "art education also". Readers are not familiar with the work like you. Please assist them to easily understand your perspective throughout the paper.

2.The use of conjunctive adverbs such as "thus", "in the light of," etc. should be used appropriately (its usage in many places were inappropriate). Example its usage in "Thus, the current study sought to shed light on........ following the sentence "Although children’s art participation is significantly dependent....."

Social class implies social groups, ranks, or levels, and so I doubt if "social class level" is appropriate.

1c. It appears you are relying solely on AI to deal with syntax and other grammatical issues raised in the previous review. The paper is still not devoid of syntax and other grammatical issues; please seek professional assistance to improve the manuscript or more work need to be done.

3.The research was conducted in 2022. Data collection spans what period?

4.Why is it hard to have a specific section in the methodology that addresses how ethical procedures were properly taken and adhered to in a study involving children? PlosOne requires detailed information on it.

5.With regard to establishing various aspects of validity,

Cite to support your values. For instance, GFI values greater than 0.95 are generally recommended. In your case, we have 0.916, 0.938, 0.942, and 0.885. (It appears something is wrong somewhere.)

Hu and Bentler, and others recommend SRMR values be between 0 to 0.08. Three of your figures were 0.113, 0.257, and 0.190. There appears to be a problem with this fit index too. (Perhaps, you can find literature to support this figures.)

You did not also establish discriminant validity or show us the loading for each item. Until these fundamental issues are resolved, it is hard to comment on the rigorousness of the analysis of a study using latent variables. You can see a statistician for support.

General Comment:

More needs to be done to improve the paper including the abstract.

7. PLOS authors have the option to publish the peer review history of their article (what does this mean?). If published, this will include your full peer review and any attached files.

Reviewer #2: No

Reviewer #3: **Yes: **Ali M. AL-Asadi

Reviewer #4: No

---

## [Author Response · Author response to Decision Letter 1]

22 Jun 2023

Cover Letter

June 23, 2023 

Dear Editors and reviewers,

I am writing this letter on behalf of my coauthor regarding our manuscript Ms. No. PONE-D-23-06961R1, entitled "The Impact of Chinese Adolescents Visual Art Participation on Self-Efficacy: A Serial Mediating Role of Cognition and Emotion."We want to express our appreciation to the respected editors and reviewers for providing us the constructive comments and suggestions to shape our manuscript for quality publication. According to our original reviewer's last review, we have obtained critical and rational minor suggestions to complete the manuscript's structure and content before the official publication. As you can see below, we have responded to each comment given by the reviewer corresponding to each page's revision has been made.

Reviewer 3 Comments

Comment 1: It is understood and expected that a literature review is part of the introduction. The heading "Literature review" should be removed, and a proper transition that leads to the heading "Art participation…" should be inserted.

Response: . The suggestion has been considered, and the changes have been made as per the advice 'Art Education' added the heading instead of literature review. 

Comment 2: Again, stating the number of males and females in the sample is customary and appropriate. Stating one but not the other expects the reader to figure it out! Please state the number of both sexes.

Response: Suggestion is considered; we added both sexes number 'with a total of 2139 students (NFemale = 1182, Nmale = 957) in grades 9 (MeanAge = 14.32, range = 13-17, SD = 0.561) participating in the study (see Table 1).'

Comment 3:When data analysis leads to "exciting" results, the reasons for such excitement should be explicitly stated. The authors highlighted the following sentences to explain why the results are exciting: "which partially negates the second hypothesis," which supports the third hypothesis," and "which validate their (it should be the) fourth hypothesis of this study." I do not believe this is informative, and again, it burdens the reader to figure out why the authors think these results are wonderful. We should always keep the readers in mind and write for them. 

Response: Thank you for the suggestion; we elaborated on our findings, and in the decision section, we added more details. 'Although a prior Chinese study stated that a mother's education and income have a stronger association with children cultural capital than the father, the present study couldn't support the argument that the target region of this study SES was moderate.' 

Comment 4: While Table 2 provides the confirmatory factor analysis results, the very few words at the end of the Data analysis procedures are hardly sufficient to address the validity of these scales. Moreover, only the fact that confirmatory factor analysis is performed to validate the hypothesized scales should be stated under the data analysis procedure. The factor analysis results should be part of the results section under its own subheading after descriptive statistics. This part should be thoroughly presented in relation to the validation of the scales. That is, explain the results found in Table 2 and reference them to Table 2.

Response: Kindly refer the revised manuscript

Comment 5: I do not believe stating chi-squared to have a highly significant value without properly stating the parameter's value is consistent with PLOS style.

Response: Suggestion is considered, and changes have made.

 Reviewer 4 Comments

Comment 1: Please consider this sentence. It seems not to be clear. Consider ensuring the free flow of understanding in that paragraph.

1b, "Furthermore, it must be stressed that besides the cognitive development of children through art education, also it strengthens the non-cognitive abilities (self-esteem, self-concept, self-efficacy,peer-relationship, emotions)," (Please replace "also it" with "art education also". Readers are not familiar with the work like you. Please assist them to easily understand your perspective throughout the paper.

2.The use of conjunctive adverbs such as "thus", "in the light of," etc. should be used appropriately (its usage in many places were inappropriate). Example its usage in "Thus, the current study sought to shed light on........ following the sentence "Although children's art participation is significantly dependent....."

Social class implies social groups, ranks, or levels, and so I doubt if "social class level" is appropriate.It appears you are relying solely on AI to deal with syntax and other grammatical issues raised in the previous review. The paper is still not devoid of syntax and other grammatical issues; please seek professional assistance to improve the manuscript or more work need to be done.

Response:. The suggestion is considered.

Comment 2: The research was conducted in 2022. Data collection spans what period?.

Response: .Thank you for the reminder, we mentioned the time spans 'sectional survey study's data was conducted in 2022 (between March – May) on junior secondary students from the rural province of Guizhou in southwestern China.'

Comment 3: Why is it hard to have a specific section in the methodology that addresses how ethical procedures were properly taken and adhered to in a study involving children? PlosOne requires detailed information on it.

Response: . The suggestion is considered; under the methodology, we added a section with 'Ethical Consideration', which provides detailed information. 

Comment 4: With regard to establishing various aspects of validity, Cite to support your values. For instance, GFI values greater than 0.95 are generally recommended. In your case, we have 0.916, 0.938, 0.942, and 0.885. (It appears something is wrong somewhere.) Hu and Bentler, and others recommend SRMR values be between 0 to 0.08. Three of your figures were 0.113, 0.257, and 0.190. There appears to be a problem with this fit index too. (Perhaps, you can find literature to support this figures.)

You did not also establish discriminant validity or show us the loading for each item. Until these fundamental issues are resolved, it is hard to comment on the rigorousness of the analysis of a study using latent variables. You can see a statistician for support.

Response: The suggestion is considered,

---

## [Editor Report · Decision Letter 2]

26 Jun 2023

The Impact of Chinese Adolescents Visual Art Participation on Self-Efficacy: A Serial Mediating Role of Cognition and Emotion

PONE-D-23-06961R2

Dear Dr. Gao,

We’re pleased to inform you that your manuscript has been judged scientifically suitable for publication and will be formally accepted for publication once it meets all outstanding technical requirements.

Kind regards,

Yuh-Yuh Li, Ph.D.

Academic Editor

PLOS ONE
---

## [Editor Report · Acceptance letter]

24 Aug 2023

PONE-D-23-06961R2 

The Impact of Chinese Adolescents Visual Art Participation on Self-Efficacy: A Serial Mediating Role of Cognition and Emotion 

Dear Dr. Gao:

I'm pleased to inform you that your manuscript has been deemed suitable for publication in PLOS ONE. Congratulations! Your manuscript is now with our production department. 

Kind regards, 

on behalf of

Dr. Yuh-Yuh Li 

Academic Editor

PLOS ONE